# Two Auxin Response Elements Fine-Tune *PINOID* Expression During Gynoecium Development in *Arabidopsis thaliana*

**DOI:** 10.3390/biom9100526

**Published:** 2019-09-25

**Authors:** André Kuhn, Bethany Runciman, William Tasker-Brown, Lars Østergaard

**Affiliations:** Department of Crop Genetics, John Innes Centre, Norwich Research Park, Norwich NR4 7UH, UK; andre.kuhn@jic.ac.uk (A.K.); b.runciman@aol.co.uk (B.R.); billy.tasker-brown@jic.ac.uk (W.T.-B.)

**Keywords:** auxin response, cis-regulation, plant development, ETTIN, PINOID, INDEHISCENT, gynoecium, gene regulation

## Abstract

The plant hormone auxin controls almost all aspects of plant development through the gene regulatory properties of auxin response factors (ARFs) which bind so-called auxin responsive elements (AuxREs) in regulatory regions of their target genes. It has been proposed that ARFs interact and cooperate with other transcription factors (TFs) to bind to complex DNA-binding sites harboring cis-elements for several TFs. Complex DNA-binding sites have not been studied systematically for ARF target genes. ETTIN (ETT; ARF3) is a key regulator of gynoecium development. Cooperatively with its interacting partner INDEHISCENT (IND), ETT regulates *PINOID* (*PID*), a gene involved in the regulation gynoecium apical development (style development). Here, we mutated two ETT-bound AuxREs within the *PID* promoter and observed increased style length in gynoecia of plants carrying mutated promoter variants. Furthermore, mutating the AuxREs led to ectopic repression of *PID* in one developmental context while leading to ectopically upregulated *PID* expression in another stage. Our data also show that IND associates with the *PID* promoter in an auxin-sensitive manner. In summary, we demonstrate that targeted mutations of cis-regulatory elements can be used to dissect the importance of single cis-regulatory elements within complex regulatory regions supporting the importance of the ETT-IND interaction for *PID* regulation. At the same time, our work also highlights the challenges of such studies, as gene regulation is highly robust, and mutations within gene regulatory regions may only display subtle phenotypes.

## 1. Introduction

Over the last decades, great progress has been made in understanding how eukaryotic transcription factors (TFs) can specifically recognise the genes that they regulate [1,2,3]. Eukaryotic TFs recognise small specific DNA sequences present in regulatory regions of their target genes, called cis-regulatory elements. Loss or mutation of these cis-regulatory elements can dramatically impact an organism’s lifestyle and development [4,5]. Specific binding to cis-regulatory elements depends on protein sequence and structure of the TF and the range of biophysical interactions that the TF properties permit [6]. In many cases, members of TF families share affinity for the same DNA motif, yet they regulate different genes [6,7,8]. The mechanisms that provide target specificity in these TF families are largely unknown, and it is likely that specific DNA binding may require additional levels of control to ensure that regulation of target genes complies with developmental needs. One way to fine-tune the specificity of a DNA-TF interaction is the ability of the TF to interact and cooperate with other DNA binding proteins. In the case that a TF complex is heterotypic and, thus, consists of TFs from different families, the number as well as the sequence of cis-regulatory elements matters for binding specificity.

Auxin response factors (ARFs) are TFs that are responsible for regulating numerous developmental processes during a plant’s lifespan in response to the plant hormone auxin (indole-3-acetic acid or IAA). Impaired response to auxin can lead to severe flower defects and low fertility (e.g., *arf6/8* and *ettin* (*ett/arf3*) mutants) or even to plant death as is the case in *monoperteros* (*mp*/*arf5*) mutants, which cannot form a root [9,10,11]. Through their B3 DNA-binding domain (DBD), ARF proteins bind to cis-regulatory elements composed of six base pairs that are called auxin responsive elements (AuxREs). The AuxRE motif has the sequence TGTCNN and was first discovered and characterised in the late 1990s in the promoter of the soybean *GH3* auxin-responsive gene [12]. Since then, it has been studied in more detail with respect to its biophysical and regulatory properties [6,13,14,15,16,17,18]. Systematically mutating AuxREs in the auxin-responsive gene *LEAFY* confirmed the physiological importance of AuxREs for auxin response [5,6]. Despite these studies, complex DNA-binding sites and TF-DNA binding have not been characterised in depth for the ARF family.

The ETT protein has been implemented in several key developmental processes including gynoecium and fruit development [10,19,20]. Recent studies on protein–protein interactions identified a diverse range of proteins that directly interacts with ETT [21,22]. These proteins include several TFs from different families [21]. One of the identified interacting partners of ETT is the bHLH-type TF INDEHISCENT (IND). IND is required for the formation of the valve margins—a tissue that allows the *Arabidopsis* fruit to open and disperse the seeds upon maturation [23]. Additionally, IND was implemented in the control of polarity and symmetry transition at the apex of the gynoecium during development. In this process, IND cooperates with the bHLH TF SPATULA (SPT) [24,25]. In both processes, IND mediates its function at least in part by controlling auxin distribution [24,25,26]. IND coordinates directional auxin flux by direct repression of the *PINOID* (*PID*) gene, which encodes a serine–threonine kinase that is fundamental for proper symmetry establishment [25,26]. *PID* phosphorylates *PIN*-*FORMED* (*PIN*) auxin efflux carriers affecting their localisation and activity and thereby mediating polar auxin transport [27,28,29].

During gynoecium development, style formation is tightly controlled by the distribution of auxin in a spatial and temporal fashion. At stage 7, auxin accumulates in two lateral foci at the apex of developing gynoecia. At stage 8, two medial auxin foci emerge. These foci fuse to form an apical auxin ring at stage 9/10 that triggers a bilateral-to-radial symmetry switch, establishing the development of the radial style [25]. The generation of this apical auxin accumulation pattern relies on transcriptional repression of *PID*. After the bilateral-to-radial symmetry switch, *PID* becomes upregulated, and the apical auxin maximum disappears at stage 12 and is absent throughout style elongation [21,25].

Characterisation of the ETT–IND interaction has shown that these two proteins control polarity at the gynoecium apex by direct regulation of *PID* gene expression. Two AuxREs, in positions −429 and −447 with respect to the start codon, were identified as ETT-binding sites [21]. Firstly, this inverted AuxRE repeat lies within a region that is evolutionarily conserved among *PID* promoters in the Brassicaceae family [21]. Secondly, it is positioned in close proximity to two G-boxes and an E-box, which can be bound by SPT and IND, respectively [24,26]. Together, this makes the locus a complex regulatory patch within the *PID* promoter. Prior studies showed that the loss of either ETT or both ETT and IND function leads to misregulation of *PID*. In addition, in vitro studies using yeast one-hybrid (Y1H) experiments showed that mutating the identified AuxRE inverted repeat leads to loss of ETT binding [21]. Together, this suggests an important role for the two AuxREs in planta. Furthermore, the extensive knowledge on the regulation of the *PID* locus makes it a suitable model for in planta studies of cis-regulatory elements and their biological relevance in the context of complex regulatory regions and more broadly on gynoecium and fruit development.

Here, we evaluate the biological importance of these two specific AuxREs for the regulation of the *PID* gene in planta. We show that the two examined AuxRE sites play a role in determining style length. Furthermore, whilst providing in planta evidence that IND binds the previously identified E-box within the *PID* promoter, we also show that this binding is sensitive to increased levels of auxin.

## 2. Materials and Methods 

### 2.1. Plant Materials

Plants were grown in soil at 22 °C in long-day conditions (16 h day/8 h dark). All transgenic plant lines, *ind-2* and the *pid-8* were in Columbia (Col-0) background.

### 2.2. Generation of Transgenic Lines

The transgenic rescue and reporter lines were generated using the Golden Gate method and transformed into *Arabidopsis* [30,31]. *PID* rescue constructs were transformed into *pid-8* mutants, the *pIND:IND:YFP* construct was transformed into *ind-2*. Other while reporter constructs were transformed into Col-0. Primers and plasmids used for cloning can be found in Appendix A.

### 2.3. Phenotypic Analysis

For the silique and style length measurements, siliques were sampled from plants when they were mature. For wild-type, ten siliques from five plants (n = 50 siliques) were collected, while for transgenics, ten siliques from five independent lines per construct were collected and photographed. Silique and style length were then measured using ImageJ 1.48 [32]. The results were statistically analysed by one-way ANOVA followed by a Tukey’s multiple comparison test using GraphPad Prism Version 5.04 (La Jolla, CA, USA, www.graphpad.com). 

### 2.4. Confocal Microscopy and Corrected Total Cell Fluorescence Quantification

Confocal images were taken using a Leica SP5 (HyD detector) confocal microscope (laser 20%, smart gain 30%, 200 hz, 8× line average, and a pinhole equivalent to 1.0× the airy disk diameter) using a 40× oil emersion lens. Excitation and detection of fluorophores were configured as follows: GFP was excited at 488 nm and detected at 498–530 nm; mCherry was excited at 561 nm and detected at 571–630 nm. For comparability the acquisition settings were set on the brightest sample and kept constant. Analysis was conducted in ImageJ 1.48 [32] using the corrected total cell fluorescence (CTCF) method [33] to quantify GFP and mCherry expression at the gynoecium apex. CTCF_GFP_-to-CTCFmCherry ratios were calculated and statistically analysed using one-way ANOVA followed by Tukey’s multiple comparison test using GraphPad Prism Version 5.04. Correlation between style length and CTCF_GFP_-to-CTCFmCherry ratios were analysed by linear regression analysis using GraphPad Prism Version 5.04.

### 2.5. Chromatin Immunoprecipitation Quantitative PCR

Transcription factor ChIP was performed in triplicate using the pIND:IND:YFP line, and data were analysed as described previously [21,34]. Immunoprecipitation (IP) was conducted using the anti-GFP antibody (Roche, Mannheim, Germany, 11814460001, Lot: 19958500), and Pierce Protein G magnetic beads (ThermoFisher, Waltham, MA, USA, 88847, Lot: SI253639). Enrichment for the E-box within the *PID* promoter was quantified using quantitative PCR (qPCR). In the ChIP_qPCR_, a region of the WUSCHEL (WUS) promoter that does not contain any E-box was used as a negative control (NC). ChIP_qPCR_ data were analysed using one-way ANOVA followed by Tukey’s multiple comparison test using GraphPad Prism Version 5.04. 

## 3. Results

### 3.1. Mutation of cis-Regulatory AuxREs within the PID Promoter Affects Style Length

To examine the relevance of the identified AuxRE sites for the regulation of *PID*, different constructs were generated and introduced into the *pid-8* mutant. Each of these constructs carries 5 kb of the *PID* promoter region upstream of the *PID* genomic sequence fused to GFP. The 5-kb promoter region either carries a wild-type form or mutated versions of the two AuxREs. In the mutated versions, the G at the second positions of the TGTCNN AuxRE sequence was mutated into a T to prevent ARF binding, as previously described (Figure 1A) [6,21]. Using this approach, single mutants in the first (−434) and second −447) AuxRE as well as a double mutant for both sites were generated (named pPID_1M_, pPID_2M_, and pPID_1+2M_, respectively). In addition, the construct also carried a nuclear-localised mCherry gene under control of a constitutive *ACTIN2* (*ACT2*) promoter. 

The *pid-8* mutant phenotype shows a dramatic reduction in fruit size and a reduction of the valves [25]. Interestingly, all four constructs could rescue the *pid-8* phenotype regarding the defects in valve development and silique length (Figure 1B–D), and this series of lines will, therefore, be referred to as ‘rescue lines’. PID has previously been shown to localise to the plasma membrane in an apolar fashion [35]. We found that this was also the case for cells at the tip of the developing gynoecium in all examined lines (Appendix A). Our macroscopic analysis of the complemented fruits suggested that the style was enlarged in lines carrying the mutated promoter variants. To test this in more detail, we measured the style lengths of the rescue lines under the microscope in comparison to wild type (*Col-0*). In five independent lines for each construct, we measured ten styles per line, and the results indeed revealed a significant difference in style length (Figure 1E,F). Specifically, whilst the constructs carrying the wild-type promoter allele (pPID_wt_) were sufficient to fully rescue the *pid-8* phenotype regarding both silique length and style length, mutating either the first (pPID_1M_) or the second (pPID_2M_) AuxRE moderately, yet significantly, increased the style length when compared to wild-type and pPID_wt_ lines. However, style lengths of the pPID_1M_ and pPID_2M_ lines did not differ significantly when compared to each other. In contrast, the double-mutant pPID_1+2M_ lines had significantly longer styles than the *Col-0*, pPID_wt_, pPID_1M_, and pPID_2M_ lines. This indicates that both AuxREs in the examined region affected style length in an additive manner. The data also suggest that the two AuxRE had an equal contribution to style length. Finally, the results imply that the two AuxRE were indeed biologically important for the correct regulation of *PID* gene expression during style development.

### 3.2. Mutating ETT-Binding AuxRE Positively Affects PID Promoter Activity at Gynoecium Stage 12

To assess whether the mutation in the AuxRE sites affected the *PID* gene expression, we decided to quantify the *PID* promoter activity in the promoter mutant lines. We generated promoter-reporter lines, that contain the same 5-kb *PID* promoter variants as the rescue lines, but this time controlled the expression of a nuclear GFP. Additionally, these constructs carried a nuclear mCherry controlled by a constitutive *ACTIN2* (*ACT2*) promoter for normalisation of the GFP signals and, thus, allowed quantitative studies of gene expression. When transformed into *Col-0* plants, the ratio of GFP to mCherry corrected total cell fluorescence (CTCF) can be used as a measure of promoter activity and gene expression [33]. The promoter activity of the different variants was examined in stage 12 gynoecia (Figure 2A–D). 

The results showed significant differences between lines of different promoter variants (Figure 2E), with weakest GFP fluorescence (CTCF_GFP_/CTCF_mCherry_) in lines carrying the wild-type promoter variant, while fluorescence in lines with the promoter mutated in the first AuxRE were moderately, but not significantly, stronger. GFP fluorescence in lines mutated in the second AuxRE was significantly increased compared to the wild-type but not compared to the single mutant in the first AuxRE. The highest relative fluorescence was measured in the doubly mutated promoter lines. For these lines, the expression was significantly higher compared to pPID_wt_ and pPID_1M_ lines but not pPID_2M_. 

Together with the style measurements, these data suggest that both the first and the second AuxRE affected *PID* promoter activity. Even though activity of pPID_1M_ lines did not differ significantly from wild-type promoter or pPID_2M_ promoter activity and appeared as intermediate between the two, combining mutations in the first and the second AuxRE had an additive effect on *PID* promoter activity. This suggests that both AuxREs are regulatory elements important for the repression of the *PID* gene. Comparing the *PID* promoter activity with the phenotypic style-length data, it becomes evident that high promoter activity at stage 12 correlated positively with longer styles (Appendix A, R2 = 0.978). Therefore, higher *PID* expression seems to promote style elongation at this stage.

### 3.3. Complex Transcription Factor Interactions Regulate PID Expression During Style Development

The presented data indicate that the two examined AuxREs affected *PID* promoter activity during style formation, as mutating any of the AuxREs increases the promoter activity significantly at stage 12. The region in which these AuxREs are found also contained other potential cis-elements. For instance, two bHLH binding motifs (G-boxes) were identified, which intriguingly overlapped with the AuxREs (Figure 3A). These G-boxes were previously shown to be candidates for binding sites for the bHLH TF, SPT [24]. In agreement with this, *PID* was ectopically expressed in *spt* mutants [25]. The IND bHLH TF has also been shown to directly bind the *PID* promoter [26], most likely via a so-called E-box positioned 57 bp downstream of the AuxRE-G-box patch (Figure 3A). The ETT–IND protein interaction is sensitive to auxin [21], however, whilst it has been demonstrated that ETT is bound to the *PID* promoter regardless of the auxin level [21], it is unknown whether this is also true for IND. To test this, ChIP_qPCR_ was carried out using an IND reporter (*pIND:IND:YFP*), which rescued the defects of the strongly indehiscent *ind-2* mutant. The ChIP experiment showed that interaction of the IND:YFP protein with the examined regulatory patch around 0.4 kb upstream of the *PID* coding region was significantly reduced after auxin treatment (Figure 3B). This implies that, in contrast to ETT, IND binds its target DNA sequence in an auxin-sensitive manner.

## 4. Discussion

Spatial and temporal control of gene expression is key for the regulation of most developmental processes. However, key regulatory genes are often subject to complex regulation that involves multiple transcription factors forming various protein complexes interacting with various cis-regulatory elements at a complex regulatory region. Often the composition of these complexes and their mode of action depends on the biological context and can differ within the same cell under different developmental stages or environmental conditions. This has complicated studies on the contribution of specific cis-regulatory elements to gene expression. 

Here, we attempted to uncover the importance of two ETT-binding AuxRE sites within the promoter of the *PID* gene for expression during *Arabidopsis* gynoecium development. *PID* is a key regulator of polar auxin transport and the main driver of apical auxin accumulation in the gynoecium, which affects style establishment and potentially style elongation [25]. During style establishment, *PID* is repressed (developmental stages 7–10). This leads to auxin accumulation at the gynoecium apex, triggering a bilateral-to-radial symmetry switch to establish the radial style [25]. At later stages, during style elongation, auxin levels decrease, and *PID* is expressed [21].

The presented data show that both AuxREs identified in the *PID* promoter play a role in the regulation of *PID* gene expression during style elongation. Phenotypic analysis showed that the *PID* gene under control of any of the four examined promoter variants was able to rescue the *pid-8* ovary defects (reduced valve development). This suggests that the examined AuxREs do not play a major role in ovary formation. Besides the two AuxREs studied here, the *PID* promoter contains several other cis-regulatory elements, including several AuxREs [21,24,26]. Although, the strongest ETT binding is associated with the two AuxREs studied here during gynoecium development, it is possible that other AuxREs are important in other developmental contexts. Further phenotypic examination, however, showed significant differences in style length between different promoter variants, suggesting that the main function of the two AuxREs is in the regulation of style development. Our data show that mutations in both AuxREs increase the promoter activity significantly at stage 12 in a dosage-dependent manner. Hence, mutating the AuxREs lead to de-repression of *PID* at developmental stage 12. 

It is well established that, besides ETT, two bHLH transcription factors, SPT and IND, are essential regulators of *PID* expression during gynoecium development [25]. In *spt* mutants as well as in *ett ind* double mutants, the *PID* gene is ectopically expressed at the apex of developing gynoecia during early stages, indicating that these factors act as repressors of *PID* expression [21,25]. Sequence analysis has identified the presence of a putative IND-binding E-box and two SPT-binding G-boxes that partially overlap with the AuxREs studied here. ETT and SPT do not appear to interact with each other, but previous studies have identified that they both can form complexes with IND [24]. Moreover, while high auxin concentrations can disrupt ETT–IND interactions, this is not the case for SPT–IND interactions [21]. These observations open the possibility that SPT and ETT may compete to bind the *PID* promoter in this region, and that the effect of changing auxin levels may depend on whether ETT or SPT is bound.

We show that IND can bind the *PID* promoter in planta, presumably via the E-box element that IND has been shown to bind in yeast one-hybrid studies [24,26]. Intriguingly, auxin treatment leads to a disruption of this IND–DNA interaction (Figure 3B), whilst ETT remains bound to the *PID* promoter in the presence of auxin [21]. The mechanism by which auxin affects IND binding to its cis-element is not yet understood. Nevertheless, it is possible that IND has an active role in recruiting ETT proteins to their binding sites in the *PID* promoter to repress gene expression in low auxin levels. High auxin concentrations can disrupt ETT–IND interactions and IND-DNA interactions, releasing IND from the *PID* promoter and leading to an up-regulation of *PID*. In contrast, data for the mutated *PID* promoter lines presented here suggest that DNA-bound ETT maintains some residual repressor activity that fine-tunes *PID* expression during style elongation.

*ETT* and *IND* are expressed at the gynoecium apex and style from early stages throughout development [21,24,26]. In contrast, *SPT* is expressed until stage 11 but absent from stage 12 and onwards [24,35]. It will be interesting to address whether this differential expression of *ETT* and *SPT* contributes to the developmentally specific regulation of *PID* expression. Alternatively, it is possible that competitive protein–DNA interactions and/or complex protein–protein interactions are important. These possibilities are not mutually exclusive and will be addressed in future studies.

## 5. Conclusions

Here, we present an approach to test the contribution of two ETT-binding AuxREs in the promoter of an auxin-responsive gene (*PID*) in the developmental context of gynoecium development. This work shows that targeted mutations of potential cis-elements can be used to dissect the importance of cis-regulation within complex regulatory regions in planta. However, it also shows the challenges involved in assessing the contributions of these elements, as gene regulation is highly robust, and mutations within gene regulatory regions may display subtle phenotypes. To conclude, this study provides a starting point for a more in-depth study understanding cis-regulation of auxin-responsive genes, as the data still do not permit full elucidation of the mechanism by which ETT regulates *PID* expression. For example, it remains an open question how ETT is recruited to its cis-binding sites and whether ETT binds as a monomer, homo-, or heterodimer. In line with this, it is yet to be addressed whether IND acts as a pioneering factor and if ETT and SPT compete to bind their overlapping DNA binding sites. Nonetheless, this work indicates that cis-elements affect the complex formation at target loci, and their regulation reveals the advantage of studying cis-regulation in planta through focusing on a specific developmental process.

## Figures and Tables

**Figure 1 biomolecules-09-00526-f001:**
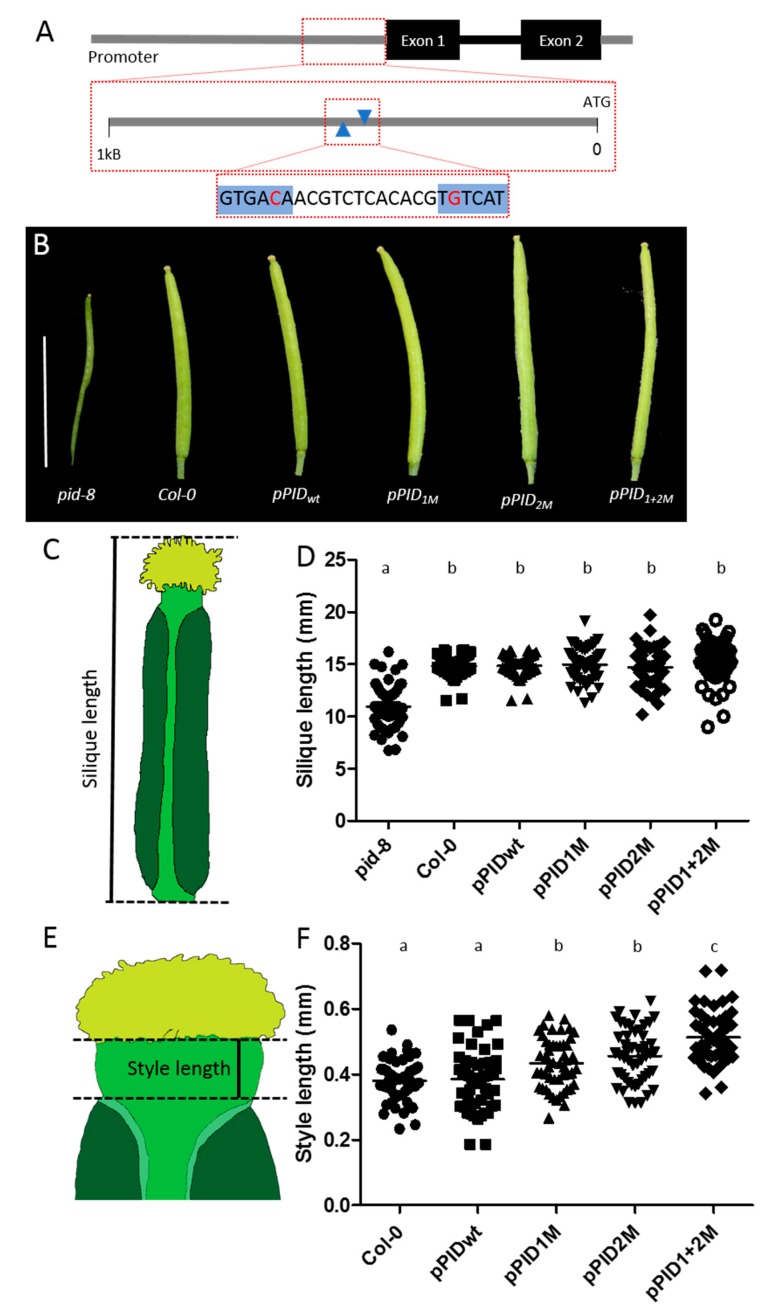
All four promoter variants can rescue the *pid-8* fruit phenotype. However, rescue lines differ significantly in style length. (**A**) Schematic representation of the regulatory region indicating the location and sequence of the two AuxREs (blue). The nucleotides that have been mutated are highlighted in red. (**B**) Representative siliques for each promoter variant construct. Scale bar = 10 mm. (**C**) Silique length describes the length of the fruit from the lower edge of the valves to the tip of the stigmatic tissue. (**D**) Silique length measurements show that all promoter variants can rescue the *pid-8* fruit length phenotype, and none of the lines differs significantly from the wild type (N = 50). (**E**) Style length is measured as the distance from the most apical part of the valves to the underside of the stigmatic tissue. (**F**) Style length measurements show that styles are significantly longer in lines with mutated AuxRE elements in the *PID* promoter compared to wild-type and the positive pPID_wt_ line (N = 50). a,b indicate significant differences according to multiple comparison.

**Figure 2 biomolecules-09-00526-f002:**
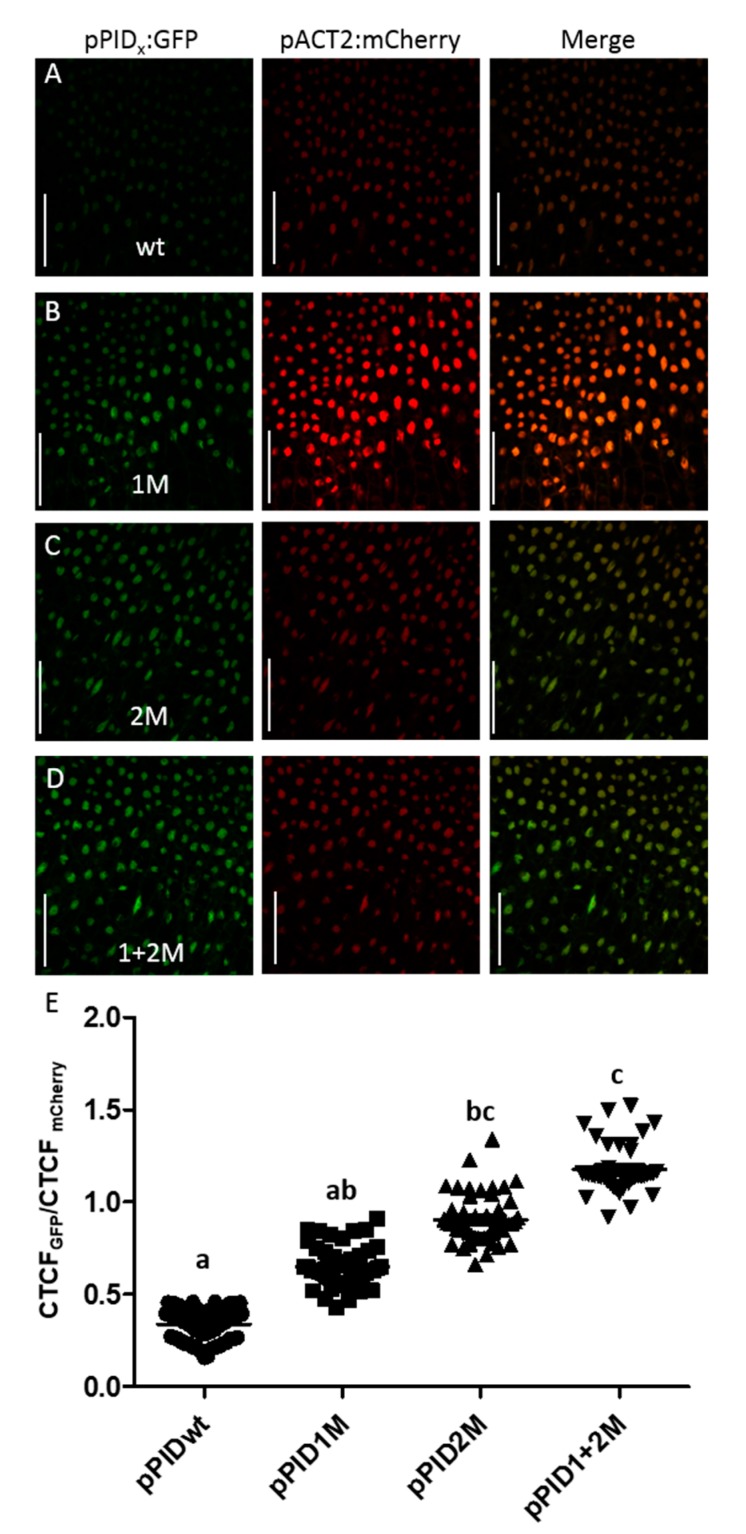
Mutating AuxREs affect *PID* promoter activity in the style region. (**A**–**D**) Representative confocal images of the style region of stage 12 for the respective promoter variant reporter lines. Scale bars = 50 µM (**E**) Fluorescence quantification shows that mutating AuxREs lead to a gradual activation of the *PID* gene. The data indicate that the two AuxREs have an additive effect (N = 40). a,b,c indicate significant differences according to multiple comparison.

**Figure 3 biomolecules-09-00526-f003:**
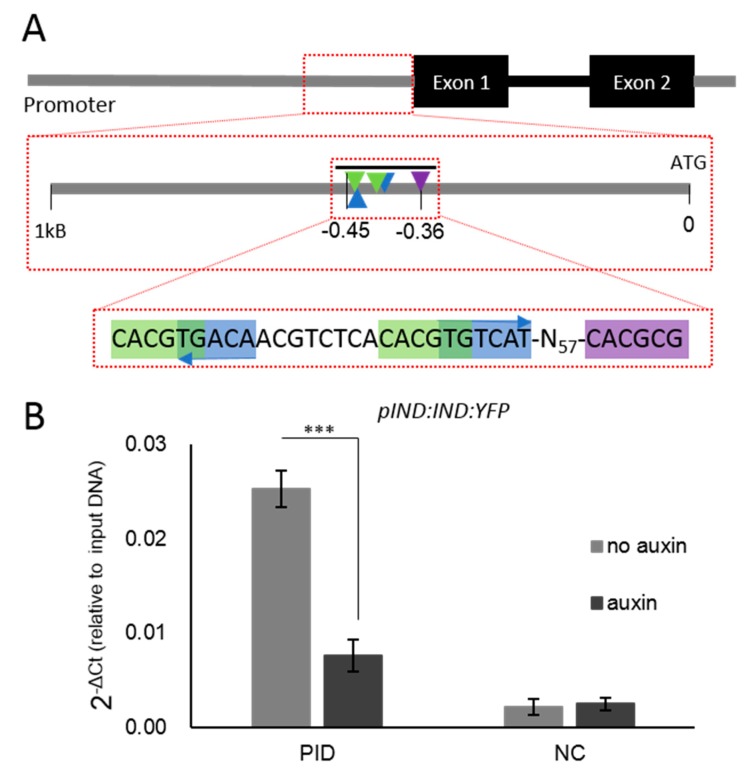
The transcription factors ETT and IND can bind to the complex regulatory patch within the *PID* promoter. (**A**) Schematic representation of the regulatory region. The black bar above the zoomed region indicates the amplicon used in qPCR reactions. Green indicates G-boxes; blue indicates AuxREs; purple indicates the E-box. (**B**) IND can bind to the regulatory patch. Auxin reduces IND binding to the *PID* promoter (*** *p*-Values < 0.0001; shown are the averages of three biological replicates ± standard deviation). NC, negative control.

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
