# Peer review of "Two Auxin Response Elements Fine-Tune PINOID Expression During Gynoecium Development in Arabidopsis thaliana"

_biomolecules, 2019, doi:10.3390/biom9100526_

Round 1

Reviewer 1 Report

The manuscript aims to investigate the importance of promoter regions involved in expression of the auxin-responsive gene PID: from the analysis of confocal microscopy, using promoter-reporter lines controlling nuclear GFP expression, and using mCherry/GFP ratio for normalization of GFP fluorescence, authors found that mutations in both AuxREs increase the promoter activity significantly at stage 12 in a dosage dependent manner. The second AuxRE appears to have the major contribution while the first AuxRE enhances repression. The work allows to further investigate the role of promoter regions, boxes, and elements, together with binding of transcription factors coactivators, in regulating the expression of target genes important in auxin signaling.

Author Response

We thank the reviewer for the positive response.

Reviewer 2 Report

The manuscript by Kuhn et al. focuses on characterization of PINOID expression regulation during gynoecium development. Authors performed mutagenesis of two auxin response elements, which are presented in the promoteric region of PINOID kinase and examined the effects of these mutations on gynoecium formation, levels of PINOID expression and transcription factors organization in this promoteric region. Overall, the manuscript is written well and the experiments performed thoroughly.

Here are some comments for the authors:

Does the mutation influence the binding of the SPT transcription factor, since the GG boxes and AuxREs are partially overlapping? Are authors sure that inability of SPT to bind to GG boxes (after mutating the AuxREs) cannot contribute to the phenotypes they observed? Also, providing some evidence that ETT cannot bind mutated AuxREs would strengthen the manuscript.

Line 159 – Authors claim that the second AuxRE have a higher contribution to style length when compared to the first AuxRE. This is not true since there is not significant difference between style length of 1M and 2M.

Figure 1F – It would be nice if authors could add also the style length of pid-8 mutant.

Line 189 – “doubly-mutated” Should be rewritten.

Line 190 – Authors claim that PINOID expression is significantly highest in the 1+2M line compared to the all other lines. It is not true as the difference in the expressions levels of 2M and 1+2M are not significantly different.

Figure 2 - Why there is such a high mCherry signal compare to the other images in the case of pPID1M? How many independent line have authors checked? Did all of the lines have elevated expression in 1M case (both PID and ACT)?

Line 202 – “de-repression” Should be substituted by activation?

Figure 3B – What does NC stands for? Should be explained at least in the figure legend.

Line 223 – “The black bar above the zoomed region indicates the amplicon used in qPCR reactions.” Cannot find anything like that in the figure.

Line 225 – “Shown are mean ± standard deviation of three biological replicates” I do not understand this statement. Should it be an average of 3 biological replicates? Please clarify.

Line 252 – “The second AuxRE appears to have the major contribution while the first AuxRE enhances repression.” I do not agree with this statement. Regarding the siliques length, style length or PINOID expression, there are no significant differences between the AuxRes, despite the mutation in the second AuxRe seems to have stronger effect. Based on the Fig. 2 pPID1M does not enhance the repression, but rather enhance the expression of PINOID. This should be rewritten.

Figure S2 – Could not find in M&M section how it was performed.

Author Response

The manuscript by Kuhn et al. focuses on characterization of PINOID expression regulation during gynoecium development. Authors performed mutagenesis of two auxin response elements, which are presented in the promoteric region of PINOID kinase and examined the effects of these mutations on gynoecium formation, levels of PINOID expression and transcription factors organization in this promoteric region. Overall, the manuscript is written well and the experiments performed thoroughly.

Here are some comments for the authors:

Does the mutation influence the binding of the SPT transcription factor, since the GG boxes and AuxREs are partially overlapping? Are authors sure that inability of SPT to bind to GG boxes (after mutating the AuxREs) cannot contribute to the phenotypes they observed?

Response: We consider it unlikely that any potentially reduced SPT binding in the mutated AuxRE lines would have any effect on the style-length phenotype studied here, because at stage 12 (the stage when style elongation occurs) SPT is absent from this tissue (Schuster et al., 2015 Development). We therefore expect that any possible effect of the mutations on SPT G-box binding would not affect style elongation. Moreover, it has been shown that the most important positions in the G-box are ACGT, which form the core. This G-box core is intact in the lines with mutated AuxREs.

Also, providing some evidence that ETT cannot bind mutated AuxREs would strengthen the manuscript.

Response: In a previous study we have shown that ETT cannot bind mutated AuxREs using a Y1H approach (Simonini et al., 2016 Genes Dev, Supplemental Figure 2c).  Additionally, using several distinct ARFs it has been shown that mutations in the second position of a core AuxRE (TGTC) leads to loss of motif binding by ARFs (Boer et al., 2014 Cell).

Line 159 – Authors claim that the second AuxRE have a higher contribution to style length when compared to the first AuxRE. This is not true since there is not significant difference between style length of 1M and 2M.

Response: The reviewer is correct, and we have addressed this comment in the revised manuscript.

Figure 1F – It would be nice if authors could add also the style length of pid-8 mutant.

Response: This is an interesting point, but it is unfortunately not straight-forward to measure the styles of pid-8. Due to its strong polarity defect the ovary of the pid-8 fruit form unevenly and as a consequence it is impossible to reliably determine the style length in these mutants.

Line 189 – “doubly-mutated” Should be rewritten.

Response: We disagree with this statement and think that “doubly-mutated” is the appropriate term to use in this context.

Line 190 – Authors claim that PINOID expression is significantly highest in the 1+2M line compared to the all other lines. It is not true as the difference in the expressions levels of 2M and 1+2M are not significantly different.

Response: Again, the reviewer is correct, and we have addressed this comment in the revised manuscript.

Figure 2 - Why there is such a high mCherry signal compare to the other images in the case of pPID1M? How many independent line have authors checked? Did all of the lines have elevated expression in 1M case (both PID and ACT)?

Response: We generated eight independent lines which showed varying levels of expression (pPID:GFP and pACT:mCherrry). For the experiments presented here we chose the four independent lines with the highest expression of pACT:mCherry. However, using the ratio between GFP and mCherry for analysis corrects for differences in the expression of the construct due to insertion localization of the T-DNA. 

Line 202 – “de-repression” Should be substituted by activation?

Response: We have substituted “de-repression” for “activation” in the revised manuscript.

Figure 3B – What does NC stands for? Should be explained at least in the figure legend.

Response: Thank you for pointing this out. We have addressed this in the Materials & Methods section and the legend.

Line 223 – “The black bar above the zoomed region indicates the amplicon used in qPCR reactions.” Cannot find anything like that in the figure.

Response: The reviewer is correct that the black bar indicating the amplicon region is missing in the figure. We have added the black bar in the figure of the revised manuscript.

Line 225 – “Shown are mean ± standard deviation of three biological replicates” I do not understand this statement. Should it be an average of 3 biological replicates? Please clarify.

Response: We agree and for clarity we have changed this to “average of three biological replicates ± standard deviation” in revised manuscript.

Line 252 – “The second AuxRE appears to have the major contribution while the first AuxRE enhances repression.” I do not agree with this statement. Regarding the siliques length, style length or PINOID expression, there are no significant differences between the AuxRes, despite the mutation in the second AuxRe seems to have stronger effect. Based on Fig. 2, pPID1M does not enhance the repression, but rather enhance the expression of PINOID. This should be rewritten.

Response: We agree with the reviewer and have modified the manuscript.

Figure S2 – Could not find in M&M section how it was performed.

Response: The reviewer is correct this information was missing in the original manuscript. It is now included in the Materials and Methods section of the revised manuscript.

Reviewer 3 Report

The manuscript by Kuhn and co-workers is well written, follows a logical order and experiments are well designed and adequate to answer the questions at hand. There is very little to comment on this manuscript, except for the few remarks listed below.

- Line 190: the double mutated promoter line is not significantly higher compared to all other lines, as the mutation in the second AuxRE element (2M line: bc) is not significantly different to the double mutated line (1+2M line: c). This should be changed in the text.

- Line 220: Although previously shown, it would be good to show both ETT and IND binding to the PID promoter in an auxin dependent or independent manner in the same experiment. Also, In Figure3, what does NC stand for (I assume a negative control, but not mentioned in the legend) and how was this region selected?

- ETT binding to PID promoter is not dependent on auxin, while IND binding is. What is the biological significance of this finding. How does this difference feed into the observed phenotypes? Can the authors elaborate on this a bit more in the discussion?

- Does the fact that even the mutated lines fully rescue the strong valve phenotype of the pidmutant show  that (likely) other regulatory cis-elements are present in the promoter, but now yet uncovered? This would be based on the very mild phenotypes observed? Can the authors discuss this point?

Author Response

The manuscript by Kuhn and co-workers is well written, follows a logical order and experiments are well designed and adequate to answer the questions at hand. There is very little to comment on this manuscript, except for the few remarks listed below.

- Line 190: the double mutated promoter line is not significantly higher compared to all other lines, as the mutation in the second AuxRE element (2M line: bc) is not significantly different to the double mutated line (1+2M line: c). This should be changed in the text.

Response: The reviewer is correct, and we have addressed this comment in the revised manuscript.

- Line 220: Although previously shown, it would be good to show both ETT and IND binding to the PID promoter in an auxin dependent or independent manner in the same experiment. Also, In Figure3, what does NC stand for (I assume a negative control, but not mentioned in the legend) and how was this region selected?

Response: The reviewer is correct the information on NC was missing in Figure 3. We have addressed this issue in the legend and the corresponding Materials and Methods section. The reviewer also suggests to show ETT binding to the PIDpromoter in presence and absence of auxin. However, given that the focus of this experiment is on identifying in vivo binding of IND to the PIDpromoter ± auxin and given that the promoter binding of ETT has been studied in detail previously (Simonini et al., 2016 Genes Dev) we do not think that this information would improve the manuscript.   

- ETT binding to PID promoter is not dependent on auxin, while IND binding is. What is the biological significance of this finding. How does this difference feed into the observed phenotypes? Can the authors elaborate on this a bit more in the discussion?

Response: This is an interesting point. In terms of ETT, this has been published in Simonini et al., 2016 Genes Dev; however, we have elaborated more on this in the Discussion section of the revised manuscript.

- Does the fact that even the mutated lines fully rescue the strong valve phenotype of the pid mutant show that (likely) other regulatory cis-elements are present in the promoter, but now yet uncovered? This would be based on the very mild phenotypes observed? Can the authors discuss this point?

Response: Thank you for this interesting comment. As already presented by our group in a previous publication (Simonini et al., 2016 Genes Dev), there are indeed several other AuxREs across the PIDpromoter; however, ChIP with ETT-GFP was strongest to the region containing the two AuxREs studied here. Even so, we have included more detail in the Discussion section of the revised manuscript.

Round 2

Reviewer 2 Report

I would like to thanks authors for considering the comments and suggestions.